# Spiraling pathways of global deep waters to the surface of the Southern Ocean

Veronica Tamsitt [1], Henri F. Drake [2,6], Adele K. Morrison[2,7], Lynne D. Talley [1], Carolina O. Dufour[2], Alison R. Gray [2], Stephen M. Griffies[3], Matthew R. Mazloff[1], Jorge L. Sarmiento[2], Jinbo Wang[4] & Wilbert Weijer[5]

Upwelling of global deep waters to the sea surface in the Southern Ocean closes the global overturning circulation and is fundamentally important for oceanic uptake of carbon and heat, nutrient resupply for sustaining oceanic biological production, and the melt rate of ice shelves. However, the exact pathways and role of topography in Southern Ocean upwelling remain largely unknown. Here we show detailed upwelling pathways in three dimensions, using hydrographic observations and particle tracking in high-resolution models. The analysis reveals that the northern-sourced deep waters enter the Antarctic Circumpolar Current via southward flow along the boundaries of the three ocean basins, before spiraling southeastward and upward through the Antarctic Circumpolar Current. Upwelling is greatly enhanced at five major topographic features, associated with vigorous mesoscale eddy activity. Deep water reaches the upper ocean predominantly south of the Antarctic Circumpolar Current, with a spatially nonuniform distribution. The timescale for half of the deep water to upwell from 30° S to the mixed layer is ~60–90 years.

---

[1] Scripps Institution of Oceanography, La Jolla, CA 92093, USA. [2] Princeton University, Princeton, NJ 08544, USA. [3] Geophysical Fluid Dynamics Laboratory, Princeton, NJ 08540, USA. [4] Jet Propulsion Laboratory, California Institute of Technology, Pasadena, CA 91109, USA. [5] Los Alamos National Laboratory, Los Alamos, NM 87545, USA. [6] Present address: Massachusetts Institute of Technology and Woods Hole Oceanographic Institution Joint Program in Oceanography, Cambridge, MA, USA. [7] Present address: Australian National University, Canberra, ACT 2602, Australia. Correspondence and requests for materials should be addressed to V.T. (email: vtamsitt@ucsd.edu)

The global overturning circulation moves waters around the world's oceans, connecting surface and deep waters through two interlinked overturning cells, one with sinking in the far northern North Atlantic and adjacent Nordic Seas and the other with sinking along the Antarctic coastline[1, 2]. These processes are well documented, with the northern sites well mapped and the southern sites, in coastal polynyas, increasingly so[3]. In contrast, the specific locations where these waters return back to the sea surface to complete the circuit are poorly known. Observations suggest that as much as 80% of the World Ocean deep water returns to the surface in the Southern Ocean with the remainder reaching the sea surface through upwelling to the thermocline in low latitudes[2, 4]. The vigor of the Southern Ocean return limb derives from the dynamics associated with the existence of an open circumpolar pathway around Antarctica in Drake Passage latitudes[5]. Dense deep water is drawn upward along steeply tilted isopycnals (surfaces of constant density), driven by divergence of wind-driven Ekman transport and surface buoyancy forcing, enabling the return of deep water to the surface with minimal diapycnal mixing[6, 7]. In the upper overturning cell, this upwelled water is transported northward via wind forcing and becomes lighter mode and intermediate waters. Below this, in the lower cell, the upwelled water is transformed into abyssal Antarctic bottom water (AABW) that sinks, moves northward, and is then converted to deep waters through diabatic mixing above the seafloor[8–10]. The warm, upwelled water that nears the ice shelves of West Antarctica[11] is recognized as a major factor in the high rate of ice shelf basal melt;[12] variability in upwelling is therefore one likely contributor to the accelerated melt rate documented in this region[13], with long-term consequences for sea level rise.

This major Southern Ocean return limb of the global overturning circulation is usually described in a two-dimensional sense (latitude-depth space), drawing on its parallel with the strongly zonally symmetric atmospheric dynamics. Mesoscale eddies have long been recognized as fundamental to the zonally averaged view of the Antarctic Circumpolar Current (ACC), arising due to baroclinic instability associated with the sharply sloped isopycnals. In the upper ocean, southward eddy-induced transport directly opposes the northward Ekman transport, limiting the residual overturning magnitude and reducing the sensitivity of the overturning to strengthening westerly winds[14, 15]. Beneath the surface layer, eddies are the primary mechanism for the southward transport of deep water across the ACC fronts[16], in the latitude and depth range that is unblocked by continental boundaries or topographic ridges ("Drake Passage effect")[1, 17]. However, recent studies have demonstrated strong zonal variations in the Southern Ocean circulation, emphasizing the importance of taking into account the three-dimensionality of the circulation[2, 16, 18–20].

The southeastward pathway that the deep waters follow, entering from the basins lying to the north and then traveling around Antarctica until reaching the continental margin, is an aspect of the Southern Ocean circulation that is familiar from maps of the surface circulation. However, this circulation is rarely explored for its interaction with the upwelling of the deep waters along this path, and for the specific locations where enhanced upwelling occurs. The ACC spirals southeastward from its northernmost latitude just east of South America to its entry into Drake Passage from the Pacific, nearly 1700 km farther south[21]. The southward shift is consistent with a vorticity balance in which mean advection of planetary vorticity by the ACC balances vorticity generation by wind stress curl (i.e., Sverdrup balance). Previous work has noted the existence of a spiral structure in the Southern Ocean upwelling[22]. However, to date the detailed geographic distribution and mechanisms for the upwelling along this ACC path have been largely unexplored.

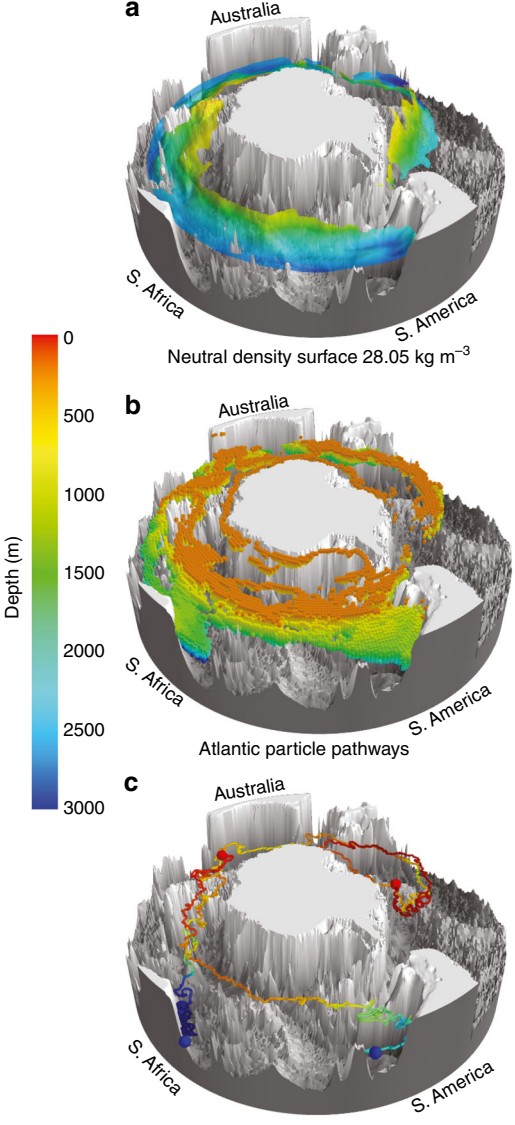

**Fig. 1** The three dimensional upward spiral of North Atlantic Deep Water through the Southern Ocean. **a** Observed warm water (>1.6 °C) on the 28.05 kg m⁻³ neutral density surface from hydrographic observations[27], south of 40° S, colored by depth (m). The isoneutral surface is masked in regions with potential temperature below 1.6 °C. 1/4° ocean bathymetry[70] is shown in *gray*. **b** Modeled (CM2.6) particle pathways from the Atlantic Ocean, with particles released in the depth range 1000–3500 m along 30° S. *Colored boxes* mark 1° latitude × 1° longitude × 100 m depth grid boxes visited by >3.5% of the total upwelling particle-transport from release at 30° S to the mixed layer. Boxes are colored by depth, similar to **a**. **c** Two example upwelling particle trajectories from CM2.6, one originating from the western Atlantic and the other from the eastern Atlantic. Trajectories are colored by depth as in **a** and **b**, *blue spheres* show the particle release locations and *red spheres* show the location where the particles reach the mixed layer. Three-dimensional maps were produced using Python and Mayavi[71]

The time scale for deep waters to reach the sea surface from each of the northern basins is important, both for setting the temporal response in the Southern Ocean to major changes in northern deep water formation rates[23] and for its control on biogeochemical processes that affect climate[24, 25]. Relatively carbon-poor North Atlantic Deep Water (NADW) mingles with much older, carbon-rich Indian and Pacific Deep Waters (IDW

and PDW) and all rise to the surface[2]. The relative amounts and time scales of these different northern deep water components impact the near-surface, upwelled ocean carbon and nutrient content and the heat supply to the Antarctic margins.

We document here the three-dimensionality of upwelling from the deep ocean interior to the surface of the Southern Ocean with observations and three independent, state-of-the-art, eddying ocean and climate models. Our analysis reveals the locations where the deep waters are most strongly shifted upwards and where they reach the sea surface. We find that upwelling along the southeastward spiral is not uniform. Where the ACC encounters major topographic features, flow-topography interactions create localized energetic eddy "hotspots"[26], which drive enhanced cross-frontal exchange[16, 19]. Here we show for the first time that deep water upwelling is also strongly enhanced at these hotspots. We also give a first estimate of the time scales of this upwelling and relative contributions of deep waters from the Atlantic, Indian, and Pacific to upwelled waters in the Southern Ocean.

## Results

**Three-dimensional deep water spiral**. The broad three-dimensional pathway of upwelling in the Southern Ocean is illustrated using observed properties[27] along a surface representing NADW (the neutral density surface 28.05 kg m$^{-3}$; Fig. 1a, Supplementary Fig. 1). The relatively warm, saline NADW, represented in Figure 1a by waters warmer than 1.6 °C, enters the Southern Ocean from the deep Atlantic (2800 m depth) and spirals southeastward and upward through the ACC. Waters warmer than 1 °C on this neutral density surface approach the Antarctic continental shelf (500 m depth) along the West Antarctic Peninsula and Amundsen Shelf south of 60° S, where incursions of upwelled, warm, northern-sourced deep waters have been implicated in the accelerated melting of ice shelves[28]. Associated maps show the separate entrances of high-nutrient/ low-oxygen IDW and PDW into the southeastward spiral (Supplementary Fig. 1 and Supplementary Note 1)[27]. The spiraling paths of NADW/IDW/PDW properties mostly follow the ACC fronts, and, upon close inspection, appear to cross fronts downstream of major topographic features (Supplementary Fig. 1).

More detailed geographic description and timescales of upwelling are difficult with the sparse Southern Ocean hydrographic data sets. We therefore use a Lagrangian modeling approach to quantify Southern Ocean upwelling and explore mechanisms controlling its pathways. We track virtual particles and their associated volume transports (particle transport) from the deep ocean interior (1000–3500 m layer) at 30° S until they reach the mixed layer in three independent eddying models: the Community Earth System Model (CESM), the Geophysical Fluid Dynamics Laboratory's Climate Model version 2.6 (CM2.6), and the Southern Ocean State Estimate (SOSE; See Methods section for model and particle tracking details). We note that while the 1000–3500 m depth range spans a broad range of deep water densities, the focus here is on interior upwelling away from bottom boundary layer processes, rather than the upwelling of AABW from the abyssal ocean.

Modeled particles from the deep Atlantic preferentially spiral southeastward and upwards through the ACC (Fig. 1b, c and Supplementary Movie 1 using CM2.6; CESM and SOSE results are qualitatively similar). Similar spirals are also clear for modeled particles released in the Pacific and Indian Oceans (Supplementary Fig. 2, Supplementary Movies 2 and 3, and Supplementary Note 2). The modeled Atlantic spiral (Fig. 1b) strongly resembles the observed pathway of the warm, saline NADW (Fig. 1a), although a different diagnostic is used (temperature on an isopycnal for the observations and probability of passing through a grid box for the

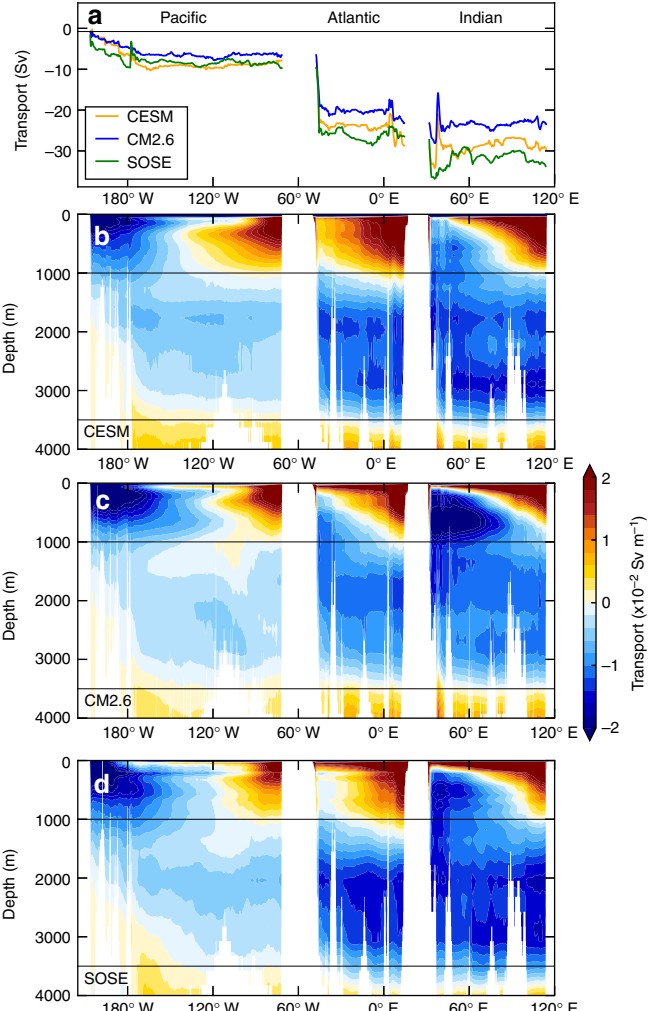

**Fig. 2** Model comparison of volume transports at 30° S. **a** Shows the volume transport in Sverdrups integrated over the depth range 1000–3500 m, **b**, **c** and **d** show the Eastward cumulative integrals of the time average meridional transport in Sv m$^{-1}$ at 30° S in CESM, CM2.6, and SOSE, respectively. The transports in **b**–**d** are normalized by the model vertical grid thicknesses

models). Additionally, the model (Fig. 1b) also shows the preferred boundary current pathways from 30° S and the near-surface continuation of the NADW pathway along the Antarctic coast, which is unclear in the NADW temperature maximum due to mixing with colder surrounding waters before reaching Antarctica.

A comparison of the time-mean volume meridional transport at 30° S in CESM, CM2.6, and SOSE shows reasonable agreement in the magnitude and spatial structure of volume transport (Fig. 2). The vertically integrated southward volume transports in the 1000–3500 m depth range agree closely in the Pacific, with the largest differences in the western Atlantic and western Indian Ocean. The total Eulerian southward transport across 30° S between 1000 and 3500 m is 28.8, 22.7, and 32.9 Sv in the CESM, CM2.6, and SOSE, respectively. These southward transports are slightly larger than the net transport in the southward limb of the zonally averaged overturning streamfunction (Fig. 2; 24.4, 21.1, and 29.0 Sv in the CESM, CM2.6, and SOSE, respectively).

By comparison, the total Lagrangian upwelling particle transport reaching the mixed layer south of 30° S is 13.2, 11.6, and 21.3 Sv in CESM, CM2.6, and SOSE, respectively. For all models, the Lagrangian transports are less than the Eulerian and overturning streamfunction transports. The two are not expected

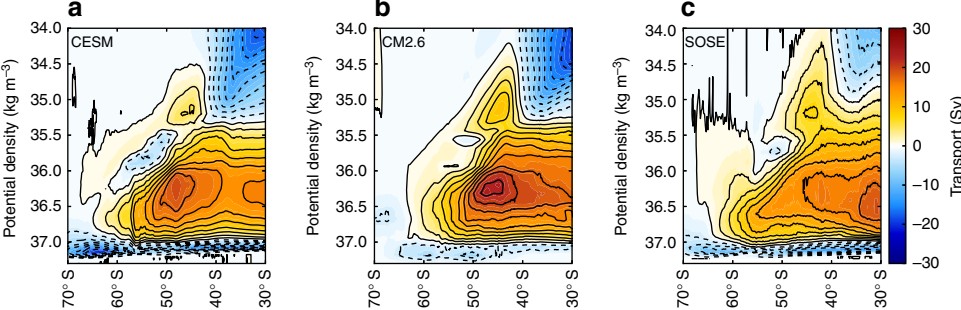

**Fig. 3** Model comparison of Southern Ocean zonally averaged circulation. Meridional overturning streamfunction (Sverdrups) in **a** CESM, **b** CM2.6, and **c** SOSE. *Solid* and *dashed* contours represent positive and negative transport, respectively, with an interval of 2.5 Sv

to agree in this case, and this is largely due to our definition of Lagrangian transport, whereby we only select particle trajectories that reach the mixed layer. In the overturning streamfunction, there is a portion of the southward upwelling limb that is entrained into either intermediate or abyssal waters in the interior without ever reaching the mixed layer. Additionally, there is likely a small fraction of Lagrangian particle-transport that takes longer than 200 years to upwell and thus is not captured in our total transport. NADW dominates the total upwelling particle-transport in all three models (51% in CM2.6 and CESM, 41% in SOSE), with the remaining transport split almost equally between the IDW and PDW.

The time taken for particles to travel from 30° S to the mixed layer is in the range of decades to more than a century, with peak upwelling occurring at 41, 28, and 81 years after release in CESM, CM2.6, and SOSE, respectively (Fig. 3a, Supplementary Table 1). We note that these transit times are considerably faster than a previous estimate of 140 years from a relatively coarse resolution (non-eddying) model[29]. We hypothesize from this previous study and our results that upwelling timescales are resolution dependent, which would explain the slower upwelling in the 1/6° SOSE compared to the 1/10° CESM and CM2.6. In CM2.6 and CESM, the median upwelling time for particle transport originating in the Indian Ocean is slightly longer than the Atlantic and Pacific, while in SOSE, particle transport from the Pacific takes substantially longer to upwell than from the Indian and Atlantic (Fig. 4b–d). There is a distinct difference in upwelling from the Indian in SOSE relative to CESM and CM2.6, with large initial upwelling in the first 25 years (Fig. 4c, *green line*). This may arise from the relatively large particle transport carried along the western boundary of the Indian Ocean by the Agulhas current in SOSE, which leads to rapid coastal upwelling from the depths in the shallower part of the 1000–3500 m range.

The three-dimensional upwelling picture (Fig. 1b) is quantified for particle trajectories from all ocean basins in a two-dimensional view (Fig. 5), revealing the horizontal pathways of upwelling and their relative strengths. Particle transport originating in the Atlantic, Indian, and Pacific Oceans at 30° S flows southward before merging into the ACC. From here, up to 20% of the total particle transport, depending on the model and basin of origin, move into parts of the Ross and Weddell Gyres and along the Antarctic coast. The pathways in Fig. 5 are remarkably insensitive to minor variations in the Lagrangian method (Supplementary Figs. 3–5 and Supplementary Note 3). There are two distinct types of pathways into the ACC: via deep western boundary currents (DWBCs) along continents or topographic ridges, and along eastern pathways whose dynamics may be eddy-driven[30]. DWBCs are the shortest and fastest routes and have been previously identified in Lagrangian experiments[31]. The DWBCs carry deep water beneath the Brazil Current in the Atlantic, beneath the Agulhas Return Current in the Indian, and

broadly below both the East Australian Current in the Tasman Sea and East Auckland Current around New Zealand and out into the deep Pacific following topography. Another deep boundary pathway in the mid-Indian Ocean follows topographic ridges, especially the Southwest Indian Ridge[32].

The eastern pathways in each ocean basin are less documented than the DWBCs. Part of the NADW leaves the Atlantic just west of South Africa, having crossed the South Atlantic at mid-latitude, consistent with both observations and models[33, 34]. This pathway is hypothesized to be driven by southward eddy thickness fluxes imposed by the northwestward movement of shallow Agulhas rings[30]. While the current has been identified in observations at 115°E by its eastward transport and low-oxygen content[35], characteristic of IDW[36], its global impact has not been appreciated and its physical cause has not been shown. We hypothesize that the eastern Atlantic eddy thickness flux mechanism[30] may also operate in the Indian, driven by eddy transport south and west of Tasmania and flowing along the southern coast of Australia[37]. In the eastern Pacific, a broad meandering pathway carries PDW southward, as identified in hydrographic observations[38, 39]. An inverse model of the Southeast Pacific circulation indicates that eddies likely play an important role in this pathway, but more work is needed to understand the underlying dynamics[39].

Although there is good agreement on the location of pathways in the three models, there are differences in the relative strengths of individual upwelling pathways. In particular, the contribution of the Pacific to the total particle transport is relatively large in SOSE, and the strength of the eastern Indian and Pacific pathways varies significantly across the models. These differences are likely attributable to differences in meridional transport at 30° S in each model (Fig. 1) or differences in model spatial and temporal resolution (see Methods section). Thus, we focus on the features that are common to all three models.

**Topographic upwelling hotspots**. Figure 5 shows the spatial distribution of particles at their final crossing of depth surfaces while upwelling. Upwelling in the ocean interior within the southeastward spiral is concentrated at the five major topographic features crossed by the ACC (shown in Fig. 6a, b for CM2.6 and Supplementary Figs. 6 and 7 for CESM and SOSE). These hotspots dominate the total upwelling across depth surfaces, with >55% of the total particle-transport upwelling across the 1000 m depth surface occurring in these five topographic hotspots in all three models, which span only 25% of the total zonal extent of the Southern Ocean (shaded in *gray* in Fig. 6a). These hotspots occur within the ACC boundaries, so that most of the upwelling across 1000 m occurs within the ACC latitude range, between 40 and 60° S (Fig. 6c). We note that there is also enhanced upwelling north of the ACC in the southward flowing western

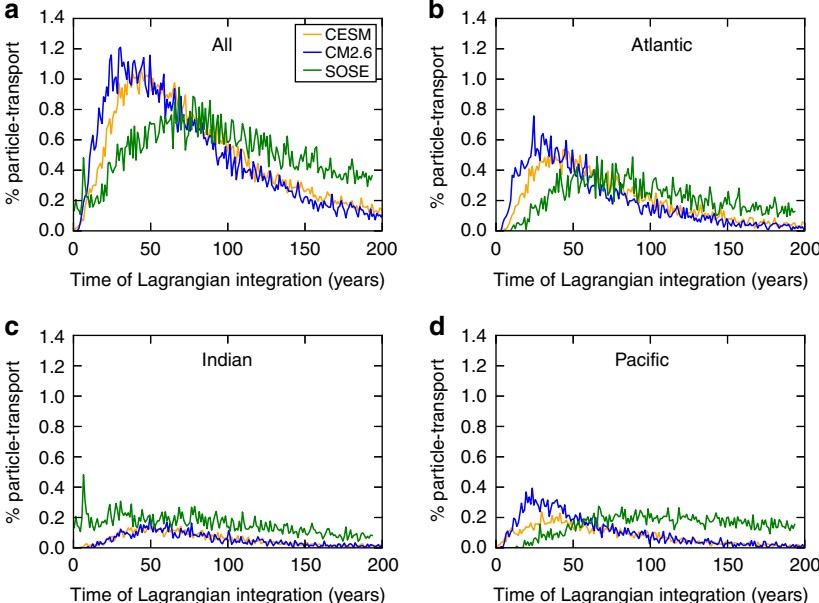

**Fig. 4** Particle upwelling timescales. Transit time distribution for particle-transport from 30° S to the mixed layer in the three models for particles originating in **a** all basins, **b** the Atlantic, **c** the Indian, and **d** the Pacific

boundary currents, but we focus our attention on the mechanism for upwelling hotspots within the ACC.

The strongly localized distribution of upwelling at 1000 m differs from the uniform upwelling expected from wind stress curl over the Southern Ocean. The hotspots of upwelling within the ACC at 1000 m occur in regions of high eddy kinetic energy (EKE, see Methods section for definition) associated with topography (*blue contours* in Fig. 6a, b, Supplementary Figs. 6 and 7, and Supplementary Note 4), where interactions between the mean flow and topography enhance eddy activity[19, 40]. Recent studies have shown preferential southward transport of particles and tracers across ACC fronts in the upper 1500 m at topographic hotspots[16, 19]. Our results show the central role of these same topographic hotspots in raising particles toward the surface as they follow the ACC path. The mean particle transport crossing 1000 m in all regions where EKE exceeds 75 cm$^2$ s$^{-2}$ is an order of magnitude larger than the mean elsewhere, and there are statistically significant correlations between mean EKE at 1000 m and particle transport crossing the 1000 m depth surface within the ACC of 0.33, 0.65, and 0.56 in CESM, CM2.6, and SOSE, respectively (Pearsons correlation coefficient with *p*-value < 0.01). Within the ACC, EKE and upwelling at 1000 m are not expected to align perfectly, because all upwelling hotspots are associated with elevated EKE, while not all regions with high EKE also have enhanced upwelling. Only locations that lie along the three-dimensional deep water pathways (Figs. 1b and 5) at the 1000 m depth surface will show enhanced upwelling.

The upwelled water in the three models reaches the surface layer, represented by upwelling across 200 m (Fig. 6d–f and Supplementary Figs. 6 and 7), mostly along the southern boundary of the ACC and over broader spatial scales than the interior upwelling hotspots. This upwelling coincides with a region of enhanced buoyancy gain by surface freshwater fluxes from melting sea ice[41]. The remaining upwelling transport reaches the surface throughout the subpolar gyres and along the Antarctic coastline, where it is exposed to buoyancy loss and may contribute to the formation of ABW. Even at 200 m, the broad distribution of upwelling, which is consistent with the broad pattern of negative wind stress curl, contains some localized enhancements associated with topographic hotspots (Fig. 6d). This agrees with a previous Lagrangian analysis that found

enhanced upwelling into the surface ocean at topographic features[42]. For example, upwelling across 200 m is enhanced in all three models at the Kerguelen Plateau, Macquarie Ridge, and Pacific–Antarctic Ridge, although there are substantial differences in the relative importance of these hotspots at 200 m between the models (Fig. 6f and Supplementary Figs. 6 and 7). These differences in particle transport at the 200 m depth surface compared to 1000 m indicates that differences in upper ocean processes between models impact the 200 m upwelling distribution, although lower spatial resolution could also contribute to the difference between SOSE and the two higher resolution models.

A schematic of a representative Southern Ocean upwelling pathway along an isopycnal surface is shown in Figure 7. Deep waters move southward from 30° S along isopycnals that are at roughly constant depth, primarily in deep boundary currents, until joining the ACC where they follow the meandering paths of the ACC fronts (*red pathway* in Fig. 7). Eddy advection drives flow across the ACC fronts in the ocean interior (*yellow arrows*). Within the ACC, isopycnals slope strongly upwards towards the surface, and simultaneously thin towards the south (Fig. 7b). Eddies act to reduce the meridional thickness gradients, hence advecting water southward and upward along isopycnals. The upwelling pathways indicate that, between topographic features, particles primarily follow mean ACC streamlines around Antarctica (Fig. 5 and schematically in Fig. 7). Where ACC fronts encounter topographic features, baroclinicity increases; strong eddy fields then develop downstream of topography[19], advecting water southwards and upwards along isopycnals. Therefore upwelling particles generally approach topographic features along more northerly ACC fronts and at greater depths, and exit downstream along more southerly ACC fronts and at shallower depths (Fig. 7b). Thus, the three-dimensional spiral is a superposition of the large-scale southeastward path of the mean ACC fronts from the Atlantic to the Pacific, and eddy-driven "steps" southward and upward across fronts at topographic hotspots. This upwelling motion along particle trajectories can be visualized as a spiral staircase.

While we propose that along-isopycnal eddy transport is the dominant mechanism for upwelling at topographic hotspots within the ACC, diapycnal mixing may also play a non-negligible role in the upwelling of deep water at these hotspots. Observations

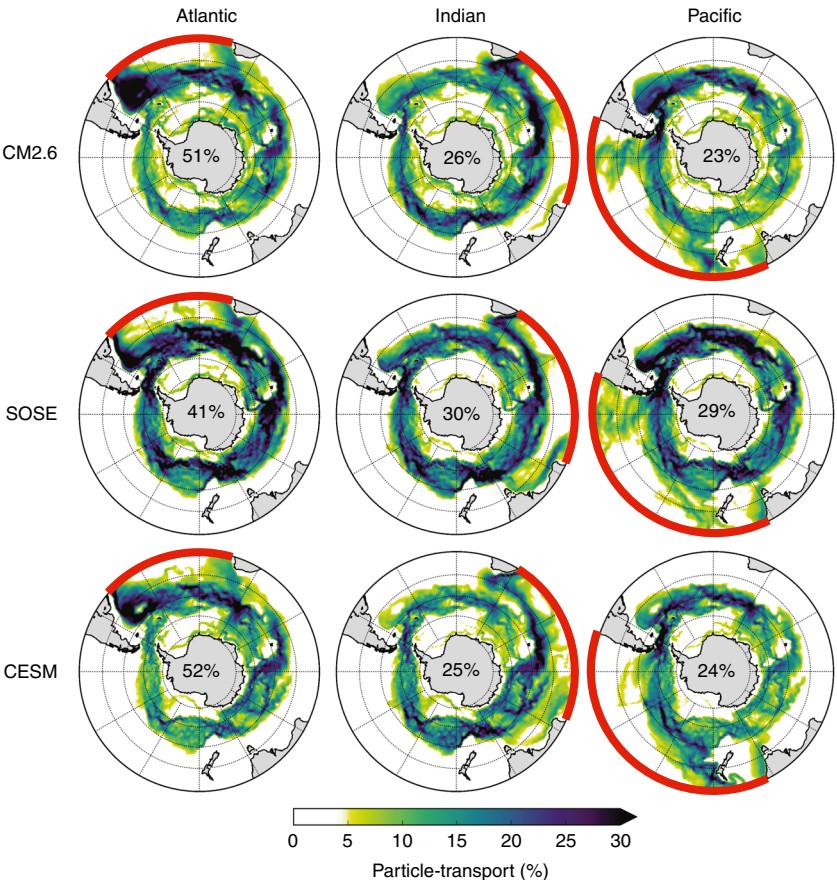

**Fig. 5** Particle pathways from 30° S to the mixed layer. Maps of the percent of total basin upwelling particle transport visiting each 1° latitude × 1° longitude grid column at some time during the 200 year experiment from release at 30° S and before reaching the surface mixed layer for CM2.6, SOSE, and CESM. The percentages of particle transports originating in the Atlantic, Indian, and Pacific (release locations at 30° S marked in *red*) are shown separately, normalized by the total upwelling particle transport originating in each basin. The percentages in the center of each panel indicate the relative contribution of the Atlantic, Indian, and Pacific to the total upwelling particle transport in each model

suggest that interior diapycnal mixing is an important component of the Southern Ocean overturning[9], [43–45], particularly in the upper 1000 m and 1000–2000 m above the seafloor[9]. Additionally, it has been shown in Drake Passage that the strength of abyssal mixing is dependent on local eddy energy[46]. In this work, the focus on upwelling at hotspots associated with enhanced eddy activity is at mid-depths away from the surface or seafloor topography. In the mid-depth ocean, along-isopycnal processes are expected to dominate over diapycnal processes. An analysis of the extent to which the interior upwelling pathways are adiabatic, and quantification of the diapycnal density change along Lagrangian trajectories at the upwelling hotspots is outside the scope of this study and is the subject of ongoing work.

## Discussion

From our results, we propose a new paradigm for the upwelling branch of the Southern Ocean overturning circulation that consists of a three-dimensional spiral, with most of the subsurface upwelling concentrated at the five major topographic features encountered by the ACC (Fig. 6): the Southwest Indian Ridge, Kerguelen Plateau, Macquarie Ridge, Pacific–Antarctic Ridge, and Drake Passage. The spatial structure of upwelling and mechanisms highlighted in this study have important implications for climate. Upwelling deep water along the Antarctic continental shelf has driven an observed acceleration in basal ice shelf melt in recent decades[28]. The three-dimensional pathways carrying deep water from the Atlantic, Indian, and Pacific to the Antarctic

continent described here provide a framework for understanding where relatively warm deep water is supplied to the Antarctic continental shelf and the origin of changes in the heat content of this water. Observations indicate that upwelling deep water preferentially reaches close to the Antarctic continent along the western Antarctic Peninsula (Fig. 1a), but further analysis of our model results are needed to determine the regionality of supply of deep water to the continental shelf in greater detail.

From our simulations, we find that the timescale for deep water in the 1000–3500 m depth range to travel from 30° S to the surface mixed layer is of the order of multiple decades to a century (Fig. 4). This upwelling timescale has implications for the time taken for changes in the deep ocean to be relayed to the surface of the Southern Ocean where they can influence the atmosphere. For instance, the peak upwelling timescale (mode) from the three models for deep water to travel from 30° S in the Atlantic Ocean to the surface of the Southern Ocean ranges from 28 to 81 years. Chlorofluorocarbon-based estimates of the timescale for water from deep water formation sites in the North Atlantic to first reach 20° S are on the order of 30 years[47]. This suggests a combined advective timescale from the northern North Atlantic to the Southern Ocean surface on the order of a century. This estimate is comparable to the time lag between abrupt climate changes in the Northern Hemisphere and Antarctica of 218 ± 92 years and 208 ± 96 years for warm and cold events, respectively, estimated from ice core records[23], which are likely propagated from the Northern Hemisphere to Antarctica via the ocean. Additionally, our estimates of Lagrangian particle transport show that NADW

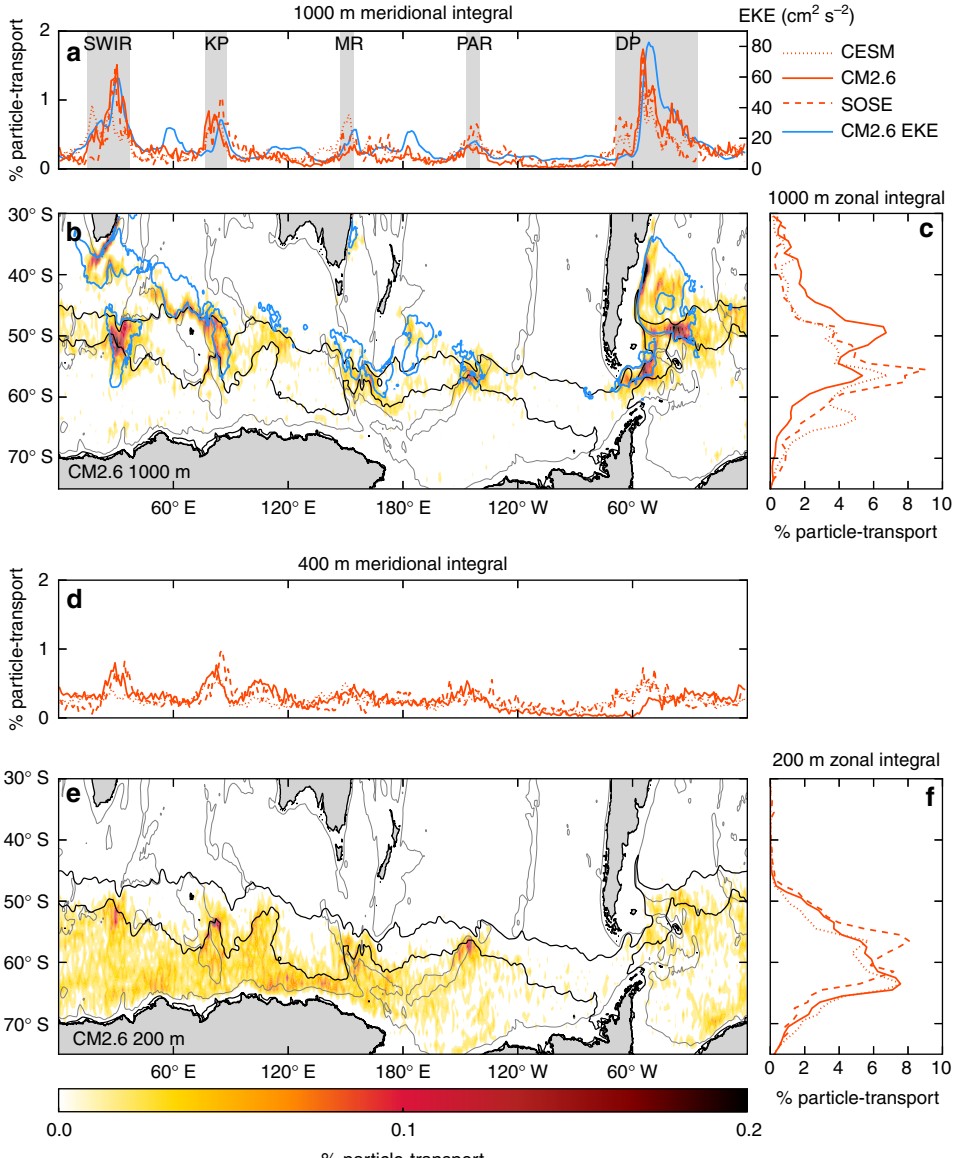

**Fig. 6** Upwelling of particles across depth horizons. **a** Percent of total upwelling particle transport crossing 1000 m (1000 m is chosen because it is representative of upwelling in the interior and lies above major topographic features) as a function of longitude, integrated across all latitudes for all three models; the *blue line* shows the mean eddy kinetic energy (EKE) at 1000 m in CM2.6 averaged between 30° S and Antarctica at each longitude; *gray shaded* bars show the location of the five major topographic upwelling hotspots: the Southwest Indian Ridge (SWIR), Kerguelen Plateau (KP), Macquarie Ridge (MR), Pacific–Antarctic Ridge (PAR), and Drake Passage (DP). **b** Percent of particle transport crossing 1000 m in each 1° latitude × 1° longitude grid box between release at 30° S and the mixed layer in CM2.6. *Blue contours* indicate regions where the mean EKE at 1000 m in CM2.6 is higher than 75 cm² s⁻². **c** Percent of particle transport crossing 1000 m depth as a function of latitude, integrated across all longitudes for all three models. **d** Same as **a** for 200 m, without EKE contours, **e** same as **b** for 200 m without EKE contours, and **f** same as **c** for 200 m. In all panels, we select the location at which particles cross depth surfaces for the final time along their trajectories. Qualitatively similar results are obtained from selecting first-crossing locations. *Black contours* in **b** and **e** are the outermost closed contours through Drake Passage of mean sea surface height in CM2.6, representing the path of the Antarctic Circumpolar Current

dominates the total upwelling. This suggests that changes in the deep Atlantic may have a disproportionate impact on the deep water properties that reach the surface of the Southern Ocean, and thus have a greater influence on heat exchange with the atmosphere and cryosphere and on delivery of warm water to the Antarctic continental shelf[48].

Our result may have ramifications for the air-sea exchange of carbon dioxide, as variability in tracer uptake in the Southern Ocean is likely related to upwelling strength[49, 50]. The spatial patterns of where deep water enriched in natural carbon but lacking in anthropogenic carbon reaches the upper ocean (Fig. 6e and Supplementary Figs. 6 and 7) are highly localized, suggesting that carbon

fluxes might also present localized patterns in relation to these upwelling hotspots, as suggested by the distribution of anthropogenic carbon uptake in an earlier iteration of SOSE[51]. Further work is needed to determine the correspondence between the distribution of upwelling into the surface ocean shown here and surface observations, and to what extent these upwelling patterns influence spatial distributions of carbon flux. The significant differences between the models in location of the deep water outcrops (Fig. 6d), in contrast with the strong agreement in the preferred locations of interior upwelling (Fig. 6a), emphasizes the importance of improving in situ observations of upwelling and carbon dioxide fluxes, which have high uncertainty due to sparse observations and large interannual

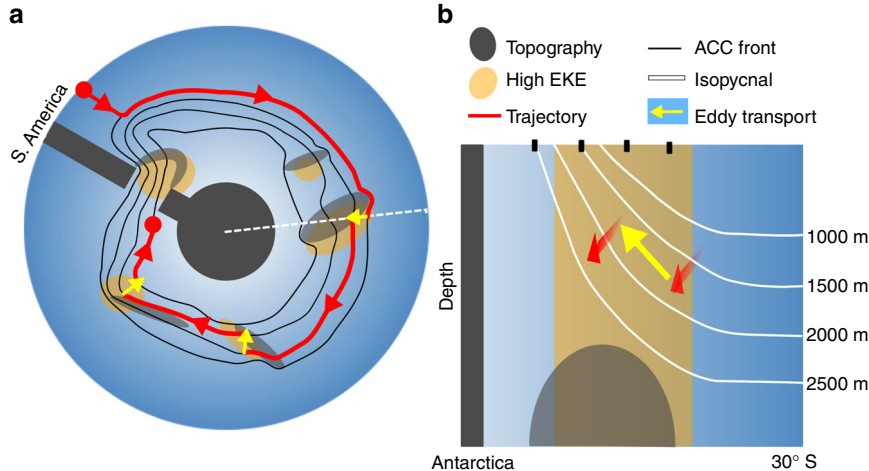

**Fig. 7** Idealized schematic illustrating the effect of eddy advection at topographic hotspots on upwelling pathways. **a** An idealized particle trajectory (*red*) follows time-mean Antarctic Circumpolar Current (ACC) streamlines (*black*) that flow southeastward around Antarctica from east of Drake Passage (*blue surface* indicating the particles' isopycnal surface, *lighter color* indicating shallower depths). The trajectory crosses streamlines and upwells (*yellow arrows*) in regions of high eddy kinetic energy (EKE; *yellow shading*) at major topographic features (*gray shading*). This creates a superimposed southward/upward spiral as the particles shift southward and upward each time they encounter a region of high EKE. **b** A two-dimensional vertical cross-section of the Southern Ocean from Antarctica to 30° S, indicated by the *white dashed line* in **a**. *White lines* show idealized isopycnal layers shoaling and thinning toward the South. The *red arrows* show the trajectory entering the high EKE region associated with topography along the northernmost ACC front and exiting the region, shallower and further south (front positions indicated by *dotted lines*)

variability[52]. The spatially varying upwelling identified here means that Southern Ocean heat and carbon uptake estimates from sparse, ship-based observations are likely unreliable. New, year-round, float-based biogeochemical measurements are beginning to transform our knowledge of the Southern Ocean carbon cycle, and will allow quantitative validation of the importance of topographic hotspots in the natural and anthropogenic carbon budgets.

Climate change is predicted to drive a strengthening in Southern Hemisphere westerly winds[53], as has already been observed in recent decades[54]. This trend has led to a more energetic eddy field in the ACC[55] and is expected to drive a further increase in EKE in the ACC in the future[56]. Our finding that eddies play a key role in driving Southern Ocean upwelling indicates that upwelling rates are likely sensitive to wind-driven changes in the eddy field. More vigorous eddies in the ACC could increase the supply of carbon-rich deep waters to the sea surface, and hence may weaken the Southern Ocean carbon sink. However, more work is needed to uncover the response of the carbon sink to a change in the eddy field. Similarly, changes in the eddy field would likely also alter the supply of nutrients to the surface of the Southern Ocean, potentially altering the efficiency of the biological pump. Our results demonstrate that a deep understanding of the three-dimensional upwelling in the Southern Ocean is needed to determine the complex role of the Southern Ocean in the global heat, carbon and nutrient budgets.

## Methods
**Observations**. Mapping of hydrographic properties on neutral density surfaces was carried out[27] using high-quality historical hydrographic data and the World Ocean Circulation Experiment (WOCE) observations of the 1990s. The maps in Figure 1a and Supplementary Figure 1 are derived from those in the WOCE Hydrographic Programme Southern Ocean Atlas[27], which used an objective mapping technique with elliptical search radii, with longer spatial scales following topographic contours. ACC fronts based on these hydrographic data are also shown in Supplementary Figure 1[21].

**Model simulations and state estimate**. Offline Lagrangian analysis was performed in two global climate models (CM2.6 and CESM) and in the regional SOSE.

CM2.6 is the high-resolution version of the Geophysical Fluid Dynamics Laboratory's CM2-O coupled model suite[57]. It combines global nominal 1/10° resolution ocean and sea ice models with 50 km resolution atmosphere and land

models. The ocean component is based on the MOM5 code, and employs no mesoscale eddy parameterization in the tracer equation. A year 1990 control simulation was used, with atmospheric $CO_2$ fixed at 355 p.p.m. CM2.6 is spun up for 84 years preceding the period used for analysis. Twelve years of 5-day averaged velocity fields were used for the Lagrangian analysis.

CESM is a high-resolution coupled climate model with nominal 1/10° ocean and sea-ice resolution and 1/4° atmosphere and land resolution[58]. The ocean component uses the Parallel Ocean Program (POP2), with no mesoscale eddy parameterizations. A year 2000 control simulation was used, with atmospheric $CO_2$ fixed at 367 p.p.m. CESM is spun up for 80 years preceding the period used for analysis here. Twenty years of monthly averaged velocity fields were used for the Lagrangian analysis.

The SOSE is a 1/6°, data-assimilating, ocean general circulation model based on the MIT General Circulation Model, configured in a domain from 24.7 to 78° S with an open northern boundary and a sea ice model[59]. No mesoscale eddy parameterization is employed. Using software developed by the consortium for Estimating the Climate and Circulation of the Ocean (http://www.ecco-group.org), the SOSE assimilates the majority of available observations using an adjoint method. For this study we used the SOSE iteration 100 solution, which has been validated against ocean and ice observations[41], and spans 6 years (2005–2010) with 1-day averaged velocity fields used for the Lagrangian analysis.

In addition to comparisons of the global model ocean and atmospheric states with observations, several papers specifically address the model representation of the ACC transport, Southern Ocean surface properties and overturning in CESM[60], CM2.6[16, 61], and SOSE[41]. A comparison of the time-mean volume meridional transport at 30° S in CESM, CM2.6, and SOSE shows reasonable agreement in the magnitude and spatial structure of volume transport (Fig. 2). The total southward transport across 30° S between 1000 and 3500 m is 28.8, 22.7, and 32.9 Sv in the CESM, CM2.6, and SOSE, respectively; the portion that does not upwell south of 30° S could be entrained into abyssal water without first reaching the sea surface, or cross north of 30° S shallower than 1000 m. Estimated total southward transport from hydrographic observations in this depth range is a comparable 18–30 Sv dependent on the choice of layer, which also include northward transport; in isopycnal layers, the maximized southward transport is order 42 Sv[62]. The Southern Ocean upper overturning cell has similar structure in the three models (Fig. 2), but the abyssal overturning cell is significantly weaker in CM2.6. The transports were calculated on potential density surfaces (referenced to 2000 m) online in CM2.6, using 30-day averaged output in CESM and on neutral density surfaces using daily averaged output in SOSE, which was remapped to approximate potential density surfaces[41].

The mixed layer depth in each model is calculated using an 0.03 kg m$^{-3}$ density threshold[63]. The upwelling pathways in all three models were found to be insensitive to the mixed layer definition (not shown). Mean EKE at 1000 m in each model was calculated from the 1-day averaged velocities in SOSE, 5-day averaged velocities in CM2.6, and 30-day averaged velocities in CESM. In this case "eddies" are defined as deviations from the long-term time-averaged velocity field.

**Lagrangian methods**. The same particle release experiment was conducted offline with velocity output from each of the three models, using the

Connectivity Modeling System[64] (CMS) in CM2.6 and CESM and Octopus (http://github.com/jinbow/Octopus) in SOSE. In each case, >2.5 million particles were released at 30° S in every grid cell between 1000 and 3500 m depth. Particles were re-released at the same location every month for the duration of the model output velocities (6 years in SOSE, 12 years in CM2.6, and 20 years in CESM). The trajectories were integrated for a total of 200 years, looping through the model output in time such that the velocity fields return to the first time step once the end of the output has been reached[65]. To avoid unphysical upwelling that might occur as a result of small model drifts when looping velocity output, the particle depths are held constant during the looping time step. The time step for particle advection was 1 h for the CMS experiments in CM2.6 and CESM, while for the Octopus experiments in SOSE, the particle advection time step was 0.5 days. A 10-min time step results in the same trajectories within a 100-day testing window because (1) the SOSE velocities are saved as daily average and (2) a high order scheme (fourth order Runge–Kutta) is used in the time integration. In Octopus, particles are numerically reflected at the sea surface and water-land boundaries. In CMS, an ad hoc boundary condition enforcing no-flux and no-slip boundary conditions is imposed; however, 30% of released particles are lost to advection into topography within 200 years. It is unlikely that this loss significantly affected the upwelling pathways, as the particles lost to topography were strongly biased toward the deepest particles with relatively low transport that were initially released near topography at 30° S. However, it is possible that this difference in handling of particles at the boundary could have contributed to the relatively large upwelling particle transport in SOSE, where no particles are lost at the boundaries.

There is no parameterization of small scale mixing used in the Lagrangian experiments, but a comparison in SOSE shows that upwelling pathways are relatively insensitive to the inclusion of a stochastic noise component to represent sub-grid scale diffusion (Supplementary Fig. 4).

After 200 years of particle advection, only particles that reached the surface mixed layer and remained south of 30° S were selected for analysis. Less than 5% of the total released particle trajectories fulfilled these criteria in all three simulations, leaving ~100,000 trajectories in each. Of the remaining 95% of particles released that did not upwell, approximately half of the particles are excluded because they had initial northward velocities and the majority of the remainder exit north of 30° S without upwelling, leaving <1.5% of particles south of 30° S that did not upwell into the mixed layer during the 200 year experiment. We only considered the portions of trajectories before particles reach the mixed layer. Pathways in Fig. 2 are insensitive to different mixed layer depth definitions or using a constant depth crossing of 200 m rather than mixed layer depth (data not shown).

It is common to use Lagrangian particle tracking to assess volume transports between chosen sites in the ocean, by assigning a volume transport to each particle at its release[31, 66–69]. Each particle was "tagged" with the meridional volume transport (in Sverdrups) at its release location at 30° S by multiplying the meridional velocity by the area of the model grid cell at the particle release location. Because the model velocity fields are non-divergent, and a sufficient number of particles are released to allow for deformation of the flow, it is assumed that the transport carried by each particle is conserved over the length of the simulation[31, 66]. This volume transport is then conserved along the trajectories until they reach the mixed layer, providing an estimate of the transport of upwelling deep water between 30° S and the mixed layer.

Particle transport weighting was used in Figs. 1 and 4–6, by summing the volume transports of each particle at each location, and normalizing by the total volume transport of all of the particles. Therefore, particles assigned with more transport initially have a larger contribution to the pathway distributions. Our spatial upwelling pathways are qualitatively unaffected by this transport weighting, although it does affect the relative timescales and strengths of different pathways. We refer to particle trajectories weighted by their initial transport at 30° S as "particle transport". The accuracy of the resulting transport pathways depends on the number of particles deployed and complexity of the flow, so the accuracy of particle-transport pathways was tested by randomly halving the selection of particles and was found to be insensitive (Supplementary Fig. 5). Other recent experiments in eddy-resolving models show good agreement between Lagrangian transports and Eulerian transports on decadal timescales[68].

**Code availability**. SOSE is based on the MITgcm code framework, available at http://mitgcm.org. Code to run the CM2.6 experiment is available from http://www.gfdl.noaa.gov/cm2-5-and-flor. The CMS, used for Lagrangian experiments with CESM and CM2.6, is an open-source Fortran toolbox available for download at https://github.com/beatrixparis/connectivity-modeling-system. The Octopus Lagrangian code, used with SOSE, is available at https://github.com/jinbow/Octopus. Analysis was completed using the open-source Python scientific stack (http://scipy.org).

**Data availability**. All data supporting this study are open and freely available. Hydrographic section data from the WOCE atlas are available at the NOAA National Centers for Environmental Information (https://www.ncei.noaa.gov/) and the CLIVAR and Carbon Hydrographic Data Office (http://cchdo.ucsd.edu). The model output from the CESM model is available through the Earth System Grid (http://earthsystemgrid.org). Output from CM2.6 used to generate figures in this paper are available from the corresponding author on reasonable request. SOSE Iteration 100 model output is available at http://sose.ucsd.edu.

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

## Acknowledgements

V.T., L.D.T., and M.R.M. were supported by NSF OCE-1357072. A.K.M., H.F.D., and W.W. were supported by the RGCM program of the US Department of Energy under Contract DE-SC0012457. J.L.S. acknowledges NSF's Southern Ocean Carbon and Climate Observations and Modeling project under NSF PLR-1425989, which partially supported L.D.T. and M.R.M. as well. C.O.D was supported by the National Aeronautics and Space Administration (NASA) under Award NNX14AL40G and by the Princeton Environmental Institute Grand Challenge initiative. A.R.G. was supported by a Climate and Global Change Postdoctoral Fellowship from the National Oceanic and Atmospheric Administration (NOAA). S.M.G. acknowledges the ongoing support of NOAA/GFDL for high-end ocean and climate-modeling activities. J.W. acknowledges support from NSF OCE-1234473 and declare that this work was done as a private venture and not in the author's capacity as an employee of the Jet Propulsion Laboratory, California Institute of Technology. Computational resources for the SOSE were provided by NSF XSEDE resource grant OCE130007. Output from the ASD run of CESM was made available by J. Small (NCAR) and colleagues. We thank Alejandro Orsi for the gridded hydrographic observations, Stu Bishop for providing processed CESM output, and Ryan Abernathey, Haidi Chen, and Nathaniel Tarshish for useful discussions on the manuscript.

## Author contributions

V.T. and H.F.D. performed Lagrangian experiments in SOSE, CM2.6, and CESM; J.W. wrote the Octopus Lagrangian code for SOSE; V.T., H.F.D., and A.K.M. analyzed model output and Lagrangian trajectories; L.D.T. analyzed the hydrographic data that motivated the model study; all authors contributed to interpretation of the results and writing of the manuscript.
