## [Peer Review File · Nature Communications]

Reviewers' comments:

Reviewer #1 (Remarks to the Author):

I think this is a very nice study, but I fail to see anything groundbreaking new. The general pathway, its spiraling structure, have been noted before. The real value of this study I see in the quantification of the pathways. On the surface, this is done very carefully, by using 3 different models and a sensitivity analysis of the Lagrangian tools.

However, I am not sure that these models are particularly well suited for that task. 2 of them are coupled models, which means their atmospheric states are possibly quite unrealistic. The other

one, SOSE, is probably the most trustworthy - for the MEAN flow. With its resolution of 1/6 degree it is not obviously suited to discuss details of eddy driven upwelling, an important aspect of the main conclusion of the study.

A basic comparison of ACC transport, wind and buoyancy flux, and SO MOC is necessary.

For the details of the Lagrangian component:

-it should be mentioned what the forward timesteps of the particle movements are.

- I 299 what does 'particle depth held constant' mean?

- I 302 if 30% are lost into topography, does this affect the overall estimates?

Lastly, I 113, the SOSE upwelling is twice that of the other 2 models. This requires more of comment than 'comparable to others'

Reviewer #2 (Remarks to the Author):

Review for Spiraling up: pathways of global deep waters to the surface of the Southern Ocean.
Tamsitt et al.

16th February 2017

This nice manuscript provides a three-dimensional view of the pathways through which the deep water masses of the Atlantic, Pacific and Indian oceans are upwelled in the Southern Ocean. Both observational data (hydrographic stations) and particle tracking models are utilized to demonstrate how water parcels assume a spiraling path upwards, and southwards within the Southern Ocean. The timescales for particles to reach the mixed-layer from 1000-3500 m depths are estimated to be of order 100 years. This study compliments other work in demonstrating that hotspots of rougher topography act to intensify both mesoscale eddy activity and upwelling at depths levels of 1000 m.

The manuscript provides important insights into the routes by which warm water is delivered to Antarctic ice-shelves and results also have implications for carbon and nutrient cycling within the global oceans. The manuscript is well written with some impressive graphics that clearly illustrate the upwelling pathways in the Southern Ocean. Hence, this study will appeal to both the expert and more general readership. I therefore recommend publication in Nature Communications, once the comments noted below have been addressed.

Finally, I should note that I am unable to verify some of the referenced papers as they are currently in review.

General comments:

There is a distinctive spatial lag between peaks in EKE and upwelling (e.g. Fig 3a). The authors do

not really discuss the reasons for the peak in EKE being downstream of the upwelling maxima (e.g. line 187). Please clarify as to how your proposed mechanism for eddy-induced upwelling fits with this spatial lag. Also I expect the meridional transport to be specific to certain depth ranges (this is eluded to in line 181). Perhaps you should clarify this.

See:

Abernathy & Cessi, Topographic enhancement of eddy efficiency in baroclinic equilibration. *J. Phys. Oceanogr.* 44, 2107-2126 (2014)

Thompson & Naveira Garabato, Equilibration of the Antarctic Circumpolar Current by standing meanders. *J. Phys. Oceanogr.* 44, 1811–1828 (2014);

Dufour (2015) as currently referenced in manuscript

In relation to this, the schematic in Figure 4b could be clarified – perhaps show a couple of panels that illustrate the different stages of how the upwelling develops. Also, as it stands, the yellow arrows implicate some kind of diapycnal mixing process.

Throughout the manuscript the authors refer to the upwelling of ‘deep’ waters. It is stated in several places that by deep waters they are referring to waters which originate from the 1000-3500 m depth in the Atlantic, Pacific and Indian oceans. However, could you clarify somewhere that this is not the same as the upwelling routes of the abyssal Antarctic Bottom Water.

How does the model distribution of upwelling correspond to surface observations?

The role of diapycnal mixing in the upwelling is rather brushed over in these studies. I think that other papers that illustrate the role of diapycnal should be mentioned [e.g. Sheen et al., Eddy induced variability in Southern Ocean abyssal mixing on climatic timescales (2014) and Naveira-Garabato et al. A microscale view of mixing and overturning across the Antarctic Circumpolar Current (2016)].

Finally, (this may be my mis-understanding) but you note in the methods that only 5% of particles released actually reached the mixed-layer and remained south of 30oS, and hence only 5% of particles are used in the manuscript analysis. How representative therefore of the Southern Ocean circulation is your analysis/videos? You should clearly state in the main article that you are only considering the 5% of water parcels that happen to show strong upwelling (and that these do so by following an upwelling spiral). How does this reconcile with the WOCE observations that appear to show an average southward flow and upwelling – indicating that the majority of particles should be following a spiraling trajectory? How might the upwelling timescales that you calculate be modified if you included other upwelling particle routes?

Minor Comments:

Line 40: also mention buoyancy forcing

Line 40: See point about what you mean by ‘deep’ waters

Lines 127-126: Any idea what percentage of particles end up in the gyres?

Lines 166: Please give significance of the correlations (are they 95% significant?). Also a correlation of 0.33 is not very large (only saying that about 10% of particle upwelling can be related to EKE in CESM). Could you try grouping the regions of highest EKE, and look to see if upwelling in these regions is significantly different?

Line 176: Please refer to figure 3f here and perhaps give an example.

Line 186: The flow speeds up - evidence?

Line 211: Please specify that you are referring to depths between 1000-3500 m here.

Line 227: Change to 'Our findings support previous studies which show that Southern Ocean upwelling is mediated by eddy activity...' with references (e.g. Gray et al.)

Figure 3 – Could you add topographic roughness (not essential I guess as key regions have been picked out in grey) and the mean latitude of upwelled particles to this plot?

Figure S1a: can you please change the colour bar limits so that changes in depth within the Southern Ocean are apparent? Also Figures S5d & e, S6c and S7c – colour bars need to be changed so that spatial distribution of anomalies can actually be seen.

Reviewer #3 (Remarks to the Author):

This paper discusses upwelling of deep water in the Southern Ocean in relation to the global overturning circulation. Focusing on upwelling pathways south of 30 degrees S, and largely relying on a Lagrangian approach to track particles in high-resolution numerical models, it provides novel evidences about the three dimensional nature of upwelling and its nonuniform spatial distribution. The subject is interesting and relevant to the understanding of the closure of the global overturning circulation and the associated climatic issues. The article is well written and, to a good extent, accessible to a broad audience including non specialists.

However, in my opinion, while the main messages are quite clear, some specific points need to be clarified.

REMARKS

(1) I particularly appreciated the use of Lagrangian analysis to investigate upwelling pathways. Nevertheless, I have some remarks about the presentation of the adopted methodology and the sensitivity of results about the modeling details.

(1a) I think that the Lagrangian modeling approach should be described with more detail to ease its understanding; only quite general features are mentioned. Alternatively, more references should be given, very few works are cited. Moreover, they are not all easily accessible. In the main text only Ref.25 is cited (p.5, line 97), which is currently under review. In the Methods section (p.15, lines 293-294) Ref.54 and 55 are cited but, again, Ref.55 is under review. Similarly, due to its importance for the presented analysis, it would be helpful to have a slightly more extended description of the volume transport method, particularly concerning the weighting. This is only briefly mentioned in Sec.2 (p.5, line 99) and in the Methods section (p.15-16, lines 314-319); again the reader is referred to Ref.25.

(1b) The looping through the model velocities (p.15, lines 297-298) used for particle trajectories' integration introduces an error associated with the jump of the velocity field. Did the Authors check its relevance? I think a comment should be added on this point.

(1c) Why are particle boundary conditions different for trajectories in different models (see p.15 lines 299-300)? Could the Authors comment on the impact of such different choices on the results?

(1d) I do not completely understand the conclusion about sensitivity to temporal averaging in

SOSE presented in the Supplementary Information (p.3 lines 84-87). The Authors say, few lines above, that previous results already indicated that higher-resolution models are insensitive to temporal averaging (using a model with comparable spatial resolution). I might be overlooking something but it is not clear to me what new information is added by the test with SOSE. Moreover, SOSE has a coarser spatial resolution and it is not evident to me how the results of this test could be transposed to CESM, which has higher spatial resolution.

(2) The results in Fig.3 (a,b) are clear and interesting. The localized enhancements mentioned on p.9 (lines 175-176), on the other hand, are more difficult to detect in Fig.3e and Fig.3d. From the last figure, differences are found between the models in the location of water outcrops. This issue is discussed on p.12 (lines 234-238) but no comment is provided about the different model resolutions, which I think could also play a role. It seems to me that the main differences are found with SOSE.

(3) In correspondence with the upwelling hotspots there should be increased vertical velocity, as one could deduce from the caption of Fig.4 (p.30, lines 519-522) and from Sec. 3 of the Supplementary Information (p.2-3 lines 54-56). I think this is an interesting point and I wonder if it could be quantified with the present data.

(4) I appreciated the discussion on time scales presented in Sec.3 of the Supplementary Information. Concerning the transit time distributions shown in Fig.S3, however, the green lines show quite different functional shapes in the Indian and Pacific with respect to the others. Do the Authors have an explanation for this? I think it would be interesting to comment on this point.

MINOR REMARKS

- p.8 lines 156-158. Does the 25% of the total zonal extent correspond to the 5 hotspots? It is not clear to me how the mentioned 55% of total particle transport can be seen in Fig.3a.

- I could not find a discussion of Fig.3c in the text.

- p.28, caption of Fig.1. How was the size of the boxes used to compute the particle pathways chosen? The Authors may compare it with the smallest scales they intend to resolve.

- I agree with Authors that the test about inclusion of diffusion (in SOSE trajectories, p.4, lines 95-100, of Supplementary Information) has several limitations in view of quantifying the role of mixing processes. Is this part really necessary? I feel that compressing it could leave more space to discuss other modeling issues as those raised above.

Moreover, no detail is provided about the stochastic term added to the particle equations of motion, except that it is isotropic. What type of noise is used here? How does the intensity of such diffusive process compare to that of the deterministic velocity in the cases considered?

- There are a couple of typos in the Supplementary Information: p.2 line 26: "and and"; p.2 line 47: "the the".

- There are no (a,b,c,d) labels in the different plots shown in Figs. S1 and S3.

Following is a point-by-point response addressing the points raised by the reviewers on the manuscript 'Spiraling up: pathways of global deep waters to the surface of the Southern Ocean. We addressed all of the comments that the reviewers made (shown in blue). We would like to thank the reviewers for the comments that improved the paper.

Reviewer #1 (Remarks to the Author):

I think this is a very nice study, but I fail to see anything groundbreaking new. The general pathway, its spiraling structure, have been noted before. The real value of this study I see in the quantification of the pathways. On the surface, this is done very carefully, by using 3 different models and a sensitivity analysis of the Lagrangian tools.

However, I am not sure that these models are particularly well suited for that task. 2 of them are coupled models, which means their atmospheric states are possibly quite unrealistic. The other one, SOSE, is probably the most trustworthy - for the MEAN flow. With its resolution of 1/6 degree it is not obviously suited to discuss details of eddy driven upwelling, an important aspect of the main conclusion of the study.

A basic comparison of ACC transport, wind and buoyancy flux, and SO MOC is necessary.

We appreciate that while previous work has noted the spiraling structure of Southern Ocean upwelling pathways, it has not been the focus of the Southern Ocean research community. This point may not have come across clearly in the manuscript, so we have edited the introduction to more explicitly acknowledge previous work, such as Döös 1995, which have noted the spiraling of the upwelling pathways (lines 61-62 in revised manuscript). However, most previous studies have looked at either the 2-dimensional zonally averaged view of the overturning (e.g. Speer et al. 2000, Lumpkin and Speer 2007, Marshall and Speer 2012), or at 2-dimensional maps showing connections between cross-frontal transport and topography (e.g. Thompson and Sallee et al. 2010, Naveira-Garabato et al. 2011, Dufour et al. 2015). In this work we are building on this prior work to provide a novel 3-dimensional view, and are moving beyond the prevailing 2-dimensional view of the Southern Ocean overturning. As far as we know, previous work has not connecting the spiral pattern to enhanced upwelling and southward transport at topographic hotspots.

Second, we believe that the differences between the model and the Lagrangian methods makes our conclusions more robust, as the spatial upwelling pathways and distribution of upwelling across the 1000 m depth surface are qualitatively similar in all three models, in spite of these differences. We acknowledge that all of these models have limitations and biases, but also that the models used are all state-of-the-art, eddying models with sufficiently high frequency velocity output that were readily available for this analysis. The atmospheric states in the coupled models, CESM and CM2.6, have been carefully compared with other models and reanalyses in detail in previous publications (Small et al. 2014 and Griffies et al. 2015). In addition, the

Southern Ocean surface properties, ACC transport and meridional overturning circulation have been compared with observations in CESM (Bishop et al. 2016) and CM2.6 (Dufour et al. 2015, Morrison et al. 2016). We have added an additional reference to these papers in the Methods section. SOSE is eddy permitting within the ACC, and while the eddy kinetic energy at 1000 m (Supplementary Figure S7a) is slightly lower in SOSE than in the higher resolution models, it exhibits a very similar spatial structure, and we believe SOSE sufficiently resolves eddies to influence upwelling within the ACC.

To aid in the ease of comparison between the three model states, we have shifted the figure showing time-mean meridional transports at 30S in each model to the main paper. We have also included an additional figure reproducing the zonal mean Southern Ocean meridional overturning circulation in each model side by side, which is described in lines 117-119, and 328-332.

For the details of the Lagrangian component:

-it should be mentioned what the forward timesteps of the particle movements are.

The forward time step for particle motion in the CM2.6 and CESM experiments using CMS are 1 hour, and in SOSE using Octopus are 0.5 days. We have added this information to the description of the Lagrangian experiment in the methods section.

- I 299 what does 'particle depth held constant' mean?

When looping velocity output, small drifts in the model could cause unphysical upwelling at the time step when the end of the model output returns to the start. To avoid this, we chose to keep the particle depths unchanged during the looping time step, thus the depth is held constant during these time steps. We have added additional explanation to clarify this (lines 345-346).

- I 302 if 30% are lost into topography, does this affect the overall estimates?

The setup for CMS used in CESM and CM2.6 imposed no-slip and no-normal-flow boundary conditions, but this resulted in 30% of particles flowing into topography. While in SOSE, we used a reflective boundary condition and no particles were lost. It is possible that this difference in boundary condition contributed to the relatively large total upwelling particle-transport in SOSE relative to the CESM and CM2.6, as no particles are lost at the boundaries in SOSE. We do not believe that the loss at topography significantly affects the upwelling transport, as these particles tend to represent deep particles released near topography that carry low transport.

Lastly, I 113, the SOSE upwelling is twice that of the other 2 models. This requires more of comment than 'comparable to others'

Indeed, the SOSE upwelling transport is larger than the other two models. A comparison of the zonally averaged Southern Ocean meridional overturning streamfunction in the three models shows that the southward component of the SOSE overturning is stronger than CESM and CM2.6, which can likely explain the larger estimate of Lagrangian upwelling transport found in SOSE. We have included additional discussion of the differences in upwelling transports in the

text (lines 115-119). Additionally, as mentioned in response to the comment above, the different Lagrangian boundary condition in SOSE compared to CESM and CM2.6 could contribute to a slight underestimate in the upwelling transport in CESM and CM2.6. We note that the Lagrangian upwelling transport as calculated here is not meant to be exactly equal to the residual overturning streamfunction, and is used as a way to determine the relative strength of upwelling pathways. Calculation of a "Lagrangian residual overturning streamfunction" using the Lagrangian trajectories, as in Döös 2008, is beyond the scope of this current work (See Döös et al. 2008 or Spence et al. 2014 for a more thorough discussion).

Reviewer #2 (Remarks to the Author):

Review for Spiraling up: pathways of global deep waters to the surface of the Southern Ocean.
Tamsitt et al.

16th February 2017

This nice manuscript provides a three-dimensional view of the pathways through which the deep water masses of the Atlantic, Pacific and Indian oceans are upwelled in the Southern Ocean. Both observational data (hydrographic stations) and particle tracking models are utilized to demonstrate how water parcels assume a spiraling path upwards, and southwards within the Southern Ocean. The timescales for particles to reach the mixed-layer from 1000-3500 m depths are estimated to be of order 100 years. This study complements other work in demonstrating that hotspots of rougher topography act to intensify both mesoscale eddy activity and upwelling at depths levels of 1000 m.

The manuscript provides important insights into the routes by which warm water is delivered to Antarctic ice-shelves and results also have implications for carbon and nutrient cycling within the global oceans. The manuscript is well written with some impressive graphics that clearly illustrate the upwelling pathways in the Southern Ocean. Hence, this study will appeal to both the expert and more general readership. I therefore recommend publication in Nature Communications, once the comments noted below have been addressed.

Finally, I should note that I am unable to verify some of the referenced papers as they are currently in review.

General comments:

There is a distinctive spatial lag between peaks in EKE and upwelling (e.g. Fig 3a). The authors do not really discuss the reasons for the peak in EKE being downstream of the upwelling maxima (e.g line 187). Please clarify as to how your proposed mechanism for eddy-induced upwelling fits with this spatial lag.

Also I expect the meridional transport to be specific to certain depth ranges (this is eluded to in line 181). Perhaps you should clarify this.

See:

Abernathy & Cessi, Topographic enhancement of eddy efficiency in baroclinic equilibration. *J. Phys. Oceanogr.* 44, 2107-2126 (2014)

Thompson & Naveira Garabato, Equilibration of the Antarctic Circumpolar Current by standing meanders. *J. Phys. Oceanogr.* 44, 1811–1828 (2014);

Dufour (2015) as currently referenced in manuscript

It is unclear that there is a consistent spatial lag between EKE and upwelling, although this is clearly the case in CESM and CM2.6 in a few specific locations. In particular, the western boundary currents (Agulhas Current and Brazil Current) show upwelling intensified on the western side of the EKE maxima. Our proposed mechanism is for upwelling within the ACC, and these southward flowing boundary currents are expected to differ from the ACC, which we have clarified in the text. In both CESM and CM2.6 there is also a noticeable upwelling hotspot immediately downstream of Kerguelen Plateau, with the EKE peak slightly further downstream. Kerguelen Plateau is the shallowest topographic obstacle to the ACC in the Drake Passage latitudes, which support a boundary current along its eastern flank (Roquet et al 2009), and thus it is likely that different dynamics may be involved in the upwelling at this specific location. In this paper, we aim to focus on the features which are common to the three models, and detailed model-specific features would require their own study which is beyond the scope of this paper. We have added a short discussion to clarify these points (lines 196-199 in revised manuscript).

With regards to the depth of eddy driven cross-frontal transport and upwelling, yes, we acknowledge that eddy transport mechanism is necessary to achieve southward cross-frontal transport in the depth range below the Ekman layer and above the sill of major topographic obstacles (~1500 m) in the Drake Passage latitudes. However, we do not think this excludes the importance of eddies outside of this depth range, and outside of the Drake Passage latitudes, where eddies continue to be important for the momentum budget, transferring momentum downward via interfacial form stress. For this reason, we have not specified a particular depth at which the eddy transport mechanism acts.

In relation to this, the schematic in Figure 4b could be clarified – perhaps show a couple of panels that illustrate the different stages of how the upwelling develops. Also, as it stands, the yellow arrows implicate some kind of diapycnal mixing process.

We acknowledge that Figure 4b could be somewhat difficult to interpret. In response, we have modified the panel to make it clearer that the motion indicated along the yellow arrows is along tilted isopycnals and does not include diapycnal processes. The fronts (marked in black) are meant to represent the surface expression of ACC fronts (e.g. sea surface height), but the southward (and upward) eddy transport indicated by the yellow arrow is along a sloping isopycnal, with no diapycnal component. We have removed the vertical black dotted lines in Figure 4b to reduce any suggestion of eddy transport across isopycnals by the yellow arrows.

Throughout the manuscript the authors refer to the upwelling of ‘deep’ waters. It is stated in several places that by deep waters they are referring to waters which originate from the 1000-3500 m depth in the Atlantic, Pacific and Indian oceans. However, could you clarify

somewhere that this is not the same as the upwelling routes of the abyssal Antarctic Bottom Water.

We acknowledge that this is an important distinction that was not made clear in the manuscript, and have included two statements to clarify (lines 39-42 in the Introduction and lines 100-102 in the Results).

How does the model distribution of upwelling correspond to surface observations?

We think it would be very valuable to compare the distribution of upwelling into the surface ocean and into the mixed layer to surface observations, particularly carbon and heat fluxes. However, significant new analysis would be required to quantify the correspondence in detail. This is beyond the scope of this study and is a subject of future work. We have modified a sentence in the discussion to reflect this (lines 275-277):

'Further work is needed to determine the correspondence between the distribution of upwelling into the surface ocean shown here and surface observations, and to what extent these upwelling patterns influence spatial distributions of anthropogenic carbon flux.'

The role of diapycnal mixing in the upwelling is rather brushed over in these studies. I think that other papers that illustrate the role of diapycnal should be mentioned [e.g. Sheen et al., Eddy induced variability in Southern Ocean abyssal mixing on climatic timescales (2014) and Naveira-Garabato et al. A microscale view of mixing and overturning across the Antarctic Circumpolar Current (2016)].

We agree that diapycnal mixing plays an important role in Southern Ocean upwelling, and this should be acknowledged more explicitly. A detailed analysis of the extent to which diapycnal processes contribute to the upwelling, and where diapycnal changes occur along upwelling particle trajectories is beyond the scope of this work, but is the subject of work currently in progress, which will soon be submitted as a separate manuscript. We have added an additional paragraph in the discussion to address this point, including the two references suggested among others (lines 230-240).

Finally, (this may be my mis-understanding) but you note in the methods that only 5% of particles released actually reached the mixed-layer and remained south of 30oS, and hence only 5% of particles are used in the manuscript analysis. How representative therefore of the Southern Ocean circulation is your analysis/videos? You should clearly state in the main article that you are only considering the 5% of water parcels that happen to show strong upwelling (and that these do so by following an upwelling spiral). How does this reconcile with the WOCE observations that appear to show an average southward flow and upwelling – indicating that the majority of particles should be following a spiraling trajectory? How might the upwelling timescales that you calculate be modified if you included other upwelling particle routes?

Although the 5% of released particles sounds like a very small fraction, we believe this captures the vast majority of the upwelling, not only 'strong upwelling' within the 200 year experiments. It is likely that a 'tail' of upwelling particles that take longer than 200 years to upwell are not captured, however the shape of the transit time distributions indicate that this tail is relatively small, particularly in CESM and CM2.6. In SOSE, where the Pacific upwelling is much slower, it

is possible that an extension of the experiment beyond 200 years would capture more upwelling in the Pacific, leading to a larger contribution to the total transport from the Pacific, but we do not expect this to impact the spatial distribution of the pathways.

Of the 95% not upwelled in the experiment, approximately half are excluded because they have initial northward velocities at release, and the vast majority of the remaining 45% cross north of 30°S without upwelling, leaving only 1.5% of particles south of 30°S that do not upwell within 200 years. We have modified the methods to include clearer and more detailed explanation of the fate of the remaining 95% of particles to make this clearer (lines 363-369).

Minor Comments:

Line 40: also mention buoyancy forcing

We have edited the sentence to include the role of buoyancy forcing in driving the overturning circulation.

Line 40: See point about what you mean by 'deep' waters

See response above. We have added an additional sentence to make the distinction between the role of diapycnal mixing in upwelling of abyssal Antarctic Bottom Water compared with upwelling of less dense deep waters (lines 39-42).

Lines 127-126: Any idea what percentage of particles end up in the gyres?

The pathway maps in Figure 5 indicate that somewhere between 5-20% of the total particle-transport enters the Ross and Weddell gyre circulation at some point during the analysis in each model. We have modified this section to reflect this (lines 146-147). However, these are concentrated along the outer boundaries of the gyre and fewer than 5% of particle-transport is found in the gyre interiors from all three basins in all three models (Figure 5). To quantify exactly how many enter anywhere in the gyres during upwelling would require defining the gyre boundaries. The distribution of upwelling across the 200 m surface (Fig. 6e, S6,S7) show that a significant percentage of particle-transport reaches the upper ocean in the subpolar gyres, particularly the eastern Weddell gyre. Again, the upwelling is concentrated along the gyre boundaries along the Antarctic continental slope and southern ACC boundary.

Lines 166: Please give significance of the correlations (are they 95% significant?). Also a correlation of 0.33 is not very large (only saying that about 10% of particle upwelling can be related to EKE in CESM). Could you try grouping the regions of highest EKE, and look to see if upwelling in these regions is significantly different?

The correlations are significant at the 99% level, this has been added to the text (line 196). It is possible that the relatively low correlation between EKE and upwelling in CESM is due to the 30 day temporal averaged velocities used for particle advection in CESM, which removes higher frequency eddy variability. We grouped the regions of high EKE and found more than 55% of the total particle-transport upwelling across the 1000 m depth surface occurring in these five

topographic hotspots in all three models, which span only 25% of the total zonal extent of the ACC (lines 179-182). Alternatively, if we take the average of upwelling across 1000 m in regions with high EKE (above $75 \text{ cm}^2\text{s}^{-1}$), compared with the average upwelling outside of these high EKE regions, the average is an order of magnitude larger in high EKE regions than outside these regions in all three models. We have added this additional result to the text. Additionally, we have added some discussion about why the correlations are not so large, and why there may be a mismatch between EKE and upwelling in some locations (lines 196-199).

Line 176: Please refer to figure 3f here and perhaps give an example.

We have added a reference to Figure 3f (now Figure 6) here and included a sentence describing examples of the topographic enhancement across all 3 models at 200 m.

Line 186: The flow speeds up - evidence?

Quasi-geostrophic modeling of jet flows along topographic ridges suggest acceleration of mean flow on the upstream side of a topographic feature, and deceleration downstream of the topographic feature (e.g. Thompson et al. 2012). Given that this point is nuanced and requires a more detailed description, and is not necessary for the explanation of enhanced upwelling in this context, we have decided to remove this phrase. The sentence now reads: 'Where ACC fronts encounter topographic features, baroclinicity increases; strong eddy fields then develop downstream of topography (Thompson et al. 2012), advecting water southwards and upwards along isopycnals.'

Line 211: Please specify that you are referring to depths between 1000-3500 m here.

The sentence has been edited to specify the depth range.

Line 227: Change to 'Our findings support previous studies which show that Southern Ocean upwelling is mediated by eddy activity...' with references (e.g. Gray et al.)

We have modified this sentence as suggested, and separated the second half of the sentence to a separate sentence for clarity.

Figure 3 – Could you add topographic roughness (not essential I guess as key regions have been picked out in grey) and the mean latitude of upwelled particles to this plot?

The latitude distribution of the upwelled particles is given in Figure 3c and f (now Figure 6), so we have added some additional explanation of this in the text. We agree that the addition of topographic roughness would be useful, but think that the topographic features have been sufficiently identified with the grey highlights in Figure 3a and 3000 m depth contour in Figure 3b and any further additions to the figure would make it cluttered and difficult to interpret.

Figure S1a: can you please change the colour bar limits so that changes in depth within the Southern Ocean are apparent?

Figure S1 is reproduced directly from the WOCE Hydrographic Programme Atlas (Orsi and Whitworth, 2005) so it is not possible to alter the colorbar limits. While we agree that the depth changes South of the ACC can be hard to discern from the colours, the colormap shows clearly

the change in depth between north of the ACC and within and south of the ACC, and the labeled contours help mark the changes to the south.

Also Figures S5d & e, S6c and S7c – colour bars need to be changed so that spatial distribution of anomalies can actually be seen.

These colour bars have been changed.

Reviewer #3 (Remarks to the Author):

This paper discusses upwelling of deep water in the Southern Ocean in relation to the global overturning circulation. Focusing on upwelling pathways south of 30 degrees S, and largely relying on a Lagrangian approach to track particles in high-resolution numerical models, it provides novel evidences about the three dimensional nature of upwelling and its nonuniform spatial distribution.

The subject is interesting and relevant to the understanding of the closure of the global overturning circulation and the associated climatic issues. The article is well written and, to a good extent, accessible to a broad audience including non specialists.

However, in my opinion, while the main messages are quite clear, some specific points need to be clarified.

REMARKS

(1) I particularly appreciated the use of Lagrangian analysis to investigate upwelling pathways. Nevertheless, I have some remarks about the presentation of the adopted methodology and the sensitivity of results about the modeling details.

(1a) I think that the Lagrangian modeling approach should be described with more detail to ease its understanding; only quite general features are mentioned. Alternatively, more references should be given, very few works are cited. Moreover, they are not all easily accessible. In the main text only Ref.25 is cited (p.5, line 97), which is currently under review. In the Methods section (p.15, lines 293-294) Ref.54 and 55 are cited but, again, Ref.55 is under review. Similarly, due to its importance for the presented analysis, it would be helpful to have a slightly more extended description of the volume transport method, particularly concerning the weighting. This is only briefly mentioned in Sec.2 (p.5, line 99) and in the Methods section (p.15-16, lines 314-319); again the reader is referred to Ref.25.

We have removed the references to Ref.25 and Ref.55 as they were anticipated to be published by the time of submission, but were not. Instead, we have included a more detailed explanation of the Lagrangian methods, particularly the volume transport method, and added additional references to multiple studies that have used this method (lines 370-387 in the revised manuscript).

(1b) The looping through the model velocities (p.15, lines 297-298) used for particle trajectories' integration introduces an error associated with the jump of the velocity field. Did the Authors check its relevance? I think a comment should be added on this point.

The models do have non-negligible drifts in the velocity and tracer fields, with larger drifts in the deep ocean than the upper ocean. We looked carefully at the magnitude of these drifts and are confident that they do not significantly affect the spatial pathways. We carefully examined the trajectories, and found there were some small but noticeable jumps when looping. To remove the possibility of unphysical upwelling due to the jump in the velocity field when looping, we added a condition that each particle is held at a constant depth during the looping timestep, and thus may change its density slightly in order to conserve its depth (this has been explained more clearly in lines 345-346). To test the sensitivity of looping in more detail would require running an additional experiment including different years of output, or rearranging the output years which would require significant computational resources that would not be possible to complete in a timely manner.

(1c) Why are particle boundary conditions different for trajectories in different models (see p.15 lines 299-300)? Could the Authors comment on the impact of such different choices on the results?

The particle boundary conditions are different purely due to the different Lagrangian particle tracking codes used. The setup for CMS used in CESM and CM2.6 imposed no-slip and no-normal-flow boundary conditions, but this resulted in 30% of particles flowing into topography. It is possible that this difference in boundary condition could contribute to the relatively large total upwelling particle-transport in SOSE relative to CESM and CM2.6, as no particles are lost at the boundaries in SOSE. However, the particles that are advected into the boundary have a strong bias toward deeper release locations, and thus tend to represent low transport. The methods section has been amended to include an explanation of this difference (lines 350-357).

(1d) I do not completely understand the conclusion about sensitivity to temporal averaging in SOSE presented in the Supplementary Information (p.3 lines 84-87). The Authors say, few lines above, that previous results already indicated that higher-resolution models are insensitive to temporal averaging (using a model with comparable spatial resolution). I might be overlooking something but it is not clear to me what new information is added by the test with SOSE.

Moreover, SOSE has a coarser spatial resolution and it is not evident to me how the results of this test could be transposed to CESM, which has higher spatial resolution.

We completed the sensitivity test to temporal averaging in SOSE because it was the model with the highest temporal resolution (daily averages) best allowing us to test the impact of averaging, and CESM output was only available as 30 day averages. Although previous results have already tested the effect of temporal averaging, we wanted to test the sensitivity for our results in particular. We have reworded this section to clarify why this test was done in SOSE.

(2) The results in Fig.3 (a,b) are clear and interesting. The localized enhancements mentioned on p.9 (lines 175-176), on the other hand, are more difficult to detect in Fig.3e and Fig.3d. From

the last figure, differences are found between the models in the location of water outcrops. This issue is discussed on p.12 (lines 234-238) but no comment is provided about the different model resolutions, which I think could also play a role. It seems to me that the main differences are found with SOSE.

Indeed there are substantial differences in the outcropping of upwelled waters at the 200 m depth surface. Most notably SOSE shows strong upwelling along the Pacific-Antarctic Ridge that is absent in the other two models. Spatial resolution could certainly play a role, but it is difficult to discern the different causes here, although there is work in progress looking at the sensitivity of the upwelling pathways to spatial resolution in the GFDL hierarchy of coupled climate models. The spatial distribution of upwelling in the upper 500 m is also dependant on mixed layer depths and upper ocean processes, which also differ between the models and arguably SOSE may be more accurately representing the spatial distribution of mixed layer processes because it is constrained to a large number of upper ocean observations. We have added additional sentences to clarify this (lines 211-214).

(3) In correspondence with the upwelling hotspots there should be increased vertical velocity, as one could deduce from the caption of Fig.4 (p.30, lines 519-522) and from Sec. 3 of the Supplementary Information (p.2-3 lines 54-56). I think this is an interesting point and I wonder if it could be quantified with the present data.

Yes, it is likely that the vertical velocity is enhanced in the upwelling hotspots. Vertical velocity is directly available from the model output, and indeed shows large (positive and negative) mean values in the upwelling hotspot regions, and was part of the motivation for investigating this upwelling with Lagrangian particles to see how these larger vertical velocities impact upwelling particle trajectories (Morrison et al. 2016). It would be possible to calculate a vertical velocity experienced by the particles, to get an 'effective' upwelling vertical velocity. However we think that this would be a considerable undertaking and beyond the scope of this current work, but could be explored in future work.

(4) I appreciated the discussion on time scales presented in Sec.3 of the Supplementary Information. Concerning the transit time distributions shown in Fig.S3, however, the green lines show quite different functional shapes in the Indian and Pacific with respect to the others. Do the Authors have an explanation for this? I think it would be interesting to comment on this point. We agree that Indian Ocean transit time distribution in SOSE (Fig S3c, green line, now Fig 4) has a different functional shape to the other two distributions in this basin. The transit time distribution from the Indian Ocean in SOSE includes significant upwelling at the beginning of the simulation, and this is due to the relatively large particle-transport carried by the Agulhas Current in SOSE, leading to particles that originate in the 1000m-2000m depth range upwelling rapidly in the western boundary current. We have added additional sentences to the timescale section discussing this (lines 138-142 in main manuscript).

In the Pacific, we think there is a less clear difference in the functional shape in SOSE relative to the other two models, but rather the peak and maximum time of the distribution are different. The times to upwell are substantially longer and more broadly distributed in SOSE, so the 200

year run does not fully capture the tail of the distribution. This difference in the peak and length of the distribution is most likely related to the lower spatial resolution of SOSE compared with CESM and CM2.6, as a separate analysis of inverse-gaussian fits to upwelling transit time distributions in the Community Modeling suite of models found that lower resolution models have broader distributions.

MINOR REMARKS

- p.8 lines 156-158. Does the 25% of the total zonal extent correspond to the 5 hotspots? It is not clear to me how the mentioned 55% of total particle transport can be seen in Fig.3a. Yes, this 25% corresponds to the 5 hotspots in Fig. 3a. The sum of the particle transport curves in Fig. 3a in the 5 hotspot regions gives 55% or a little more of the total particle transport in each model. This has been reworded to clarify.

- I could not find a discussion of Fig.3c in the text.
We have added a sentence in the explanation of Figure 3 to describe panel c.

- p.28, caption of Fig.1. How was the size of the boxes used to compute the particle pathways chosen? The Authors may compare it with the smallest scales they intend to resolve. The 1 degree longitude by 1 degree latitude boxes were chosen for computational efficiency, and a test with smaller boxes (1/2 degree) in SOSE did not yield a visibly different result.

- I agree with Authors that the test about inclusion of diffusion (in SOSE trajectories, p.4, lines 95-100, of Supplementary Information) has several limitations in view of quantifying the role of mixing processes. Is this part really necessary? I feel that compressing it could leave more space to discuss other modeling issues as those raised above. Moreover, no detail is provided about the stochastic term added to the particle equations of motion, except that it is isotropic. What type of noise is used here? How does the intensity of such diffusive process compare to that of the deterministic velocity in the cases considered? We agree there are limitations in the inclusion of diffusion in particle trajectories to quantify mixing. However, currently this is the standard way of representing mixing in Lagrangian experiments, so we felt it was important to show that its inclusion does not significantly affect our results. We have included further specific details of the stochastic term (Supplementary Information lines 61-67).

- There are a couple of typos in the Supplementary Information: p.2 line 26: "and and"; p.2 line 47: "the the".
Corrected.

- There are no (a,b,c,d) labels in the different plots shown in Figs. S1 and S3.
Corrected.

REVIEWERS' COMMENTS:

Reviewer #1 (Remarks to the Author):

Thank you for carefully addressing my concerns.

Reviewer #2 (Remarks to the Author):

I am satisfied that the majority of points which I raised in the previous review have been adequately addressed.

I fail to see any change in the schematic (figure 7), but I would not see this as a show-stopper to prevent publication.

Regards the comparison to observations - the authors have responded to say that this is outside the scope of the paper, and have now eluded to the need of this comparison in future studies.

The conclusions are still robust within the context of the three models and results they show - of course to be sure these hold in the 'real' world, observations are required. However, as the authors suggest, this should perhaps be left to a further study.

Many thanks

Reviewer #3 (Remarks to the Author):

The authors answered my questions in a thorough way. I have a few minor remarks on this revised paper, as listed below. I would recommend publication after addressing these points.

(1) p.8 line 193. I think the units of EKE should be cm^2s^{-2} here.

(2) p.8 line 196. Please specify the meaning of p in the parenthesis ($p < 0.01$).

(3) p.8-9 lines 200-214, Fig.6 (and caption). Panels d, e, f are different from those of the corresponding figure (Fig.3) of the previous version of the manuscript. Moreover, labels in panels d and e refer to 400 m, while the text as well as the related further material reported in the Supplementary Information refer to upwelling across 200 m. Figure 6 (panels d, e, f) should then be replaced by its previous version, referring to 200 m. Alternatively, the related text should be modified accordingly.

(4) p.12 line 298. The title "Observations" (in bold) should be removed, since the corresponding text has been removed.

(5) Caption of Fig.6. "... (b) and (c) are the outermost...". Shouldn't (c) be (e)?

(6) About the tests on the sensitivity to temporal averaging of velocities, I still think that the test with SOSE is not particularly compelling to infer that particles advected by CESM velocity fields are not affected by averaging. In my opinion, the reason why the authors' conclusion might apply is that they consider 1-particle statistics only. Indeed, the latter can be expected to be mainly affected by the largest and most energetic flow structures, which shouldn't be very different in the models they consider. This could probably be mentioned.

(7) The numbering of figures in the Supplementary Information should be revised. In particular S4 should be S2 (on p.2), S5 should be S3, S7 should be S5, S8 should be S6 and S9 should be S7.

(8) p.3 line 66 (Supplementary Information). Instead of saying "using a random number generator", the authors could provide a short description of the properties of the random numbers they use. Considering the subsequent discussion, I guess these have zero mean and unit variance.

(9) p.3 line 67 (Supplementary Information). A justification of the values of horizontal/vertical diffusivity used should be provided.

(10) p. 3 line 72 (Supplementary Information). In what sense "isotropic"? The stochastic displacement you add is characterized by a horizontal diffusivity that is much larger than the vertical one. You probably mean that this type of movement is "unbiased", in the sense that, for each direction, there is no drift and positive and negative displacements are equally probable.

Following is a point-by-point response addressing the points raised by the reviewers in the second round of revisions on the manuscript 'Spiraling pathways of global deep waters to the surface of the Southern Ocean'. We addressed all of the comments that the reviewers made (shown in blue). We would like to thank the reviewers for the comments that improved the paper.

Reviewer #1 (Remarks to the Author):

Thank you for carefully addressing my concerns.

Reviewer #2 (Remarks to the Author):

I am satisfied that the majority of points which I raised in the previous review have been adequately addressed.

I fail to see any change in the schematic (figure 7), but I would not see this as a show-stopper to prevent publication.

Regards the comparison to observations - the authors have responded to say that this is outside the scope of the paper, and have now eluded to the need of this comparison in future studies.

The conclusions are still robust within the context of the three models and results they show - of course to be sure these hold in the 'real' world, observations are required. However, as the authors suggest, this should perhaps be left to a further study.

Many thanks

Reviewer #3 (Remarks to the Author):

The authors answered my questions in a thorough way. I have a few minor remarks on this revised paper, as listed below. I would recommend publication after addressing these points.

(1) p.8 line 193. I think the units of EKE should be cm^2s^{-2} here.
Corrected.

(2) p.8 line 196. Please specify the meaning of p in the parenthesis ($p < 0.01$).
This has been clarified.

(3) p.8-9 lines 200-214, Fig.6 (and caption). Panels d, e, f are different from those of the corresponding figure (Fig.3) of the previous version of the manuscript. Moreover, labels in panels d and e refer to 400 m, while the text as well as the related further material reported in the Supplementary Information refer to upwelling across 200 m. Figure 6 (panels d, e, f) should then be replaced by its previous version, referring to 200 m. Alternatively, the related text should be modified accordingly.

Indeed, panels d), e) and f) in the revised manuscript showed 400 m upwelling instead of 200 m, as described previously. This has been updated with the correct figures and labels from the original manuscript.

(4) p.12 line 298. The title "Observations" (in bold) should be removed, since the corresponding text has been removed.

The paragraph below this subtitle was deleted by mistake in the revised manuscript, so has been replaced, unmodified from the original manuscript. We apologize for confusion.

(5) Caption of Fig.6. "... (b) and (c) are the outermost...". Shouldn't (c) be (e)?

Yes, this has been corrected.

(6) About the tests on the sensitivity to temporal averaging of velocities, I still think that the test with SOSE is not particularly compelling to infer that particles advected by CESM velocity fields are not affected by averaging. In my opinion, the reason why the authors' conclusion might apply is that they consider 1-particle statistics only. Indeed, the latter can be expected to be mainly affected by the largest and most energetic flow structures, which shouldn't be very different in the models they consider. This could probably be mentioned.

This comment raises an important point. As suggested, we have include the following statement explaining that the use of single particle statistics could potentially explain the similarity between the model results: 'It is important to note that only single-particle statistics are used in this analysis, which are expected to be mainly affected by the most energetic parts of the flow and thus are less sensitive to sampling frequency than other Lagrangian statistics. This could explain why the CESM results are similar to CM2.6 and SOSE even though the mesoscale is not well sampled in CESM.'

(7) The numbering of figures in the Supplementary Information should be revised. In particular S4 should be S2 (on p.2), S5 should be S3, S7 should be S5, S8 should be S6 and S9 should be S7.

The figure numbers in the Supplementary Information have been updated to refer to the correct figures.

(8) p.3 line 66 (Supplementary Information). Instead of saying "using a random number generator", the authors could provide a short description of the properties of the random numbers they use. Considering the subsequent discussion, I guess these have zero mean and unit variance.

We have included a more detailed description of the random number generator, including a reference to Kinderman and Monahan 1977.

(9) p.3 line 67 (Supplementary Information). A justification of the values of horizontal/vertical diffusivity used should be provided.

The horizontal and vertical diffusivities are chosen to be similar to the explicit diffusivities used in the Southern Ocean State Estimate (Mazloff et al. 2010), which were chosen based on the horizontal and vertical resolution of the Southern Ocean State Estimate setup. The $25 \text{ cm}^2/\text{s}$ horizontal diffusivity is first derived according the scaling of a hyperdiffusivity. There is no theoretical equivalence of hyperdiffusivity in Brownian diffusion, so it is scaled according to the length scale, which is 10^4 . We have included a sentence to reflect this.

(10) p. 3 line 72 (Supplementary Information). In what sense "isotropic"? The stochastic displacement you add is characterized by a horizontal diffusivity that is much larger than the vertical one. You probably mean that this type of movement is "unbiased", in the sense that, for each direction, there is no drift and positive and negative displacements are equally probable. This sentence has been edited to clarify. It now reads: 'First, the stochastic noise we added to the trajectories is unbiased (positive and negative displacements are equally probable), which we know is not true in reality.'